# Leveraging large multi-center cohorts of Alzheimer disease endophenotypes to understand the role of Klotho heterozygosity on disease risk

Muhammad Ali[1,2], Yun Ju Sung[1,2], Fengxian Wang[1,2], Maria V. Fernández[1,2], John C. Morris[3,4], Anne M. Fagan[3,4], Kaj Blennow[5,6], Henrik Zetterberg[5,6,7,8], Amanda Heslegrave[7,8], Per M. Johansson[7,8,9,10], Johan Svensson[11], Bengt Nellgård[11], Alberto Lleó[12], Daniel Alcolea[12], Jordi Clarimon[12], Lorena Rami[13], José Luis Molinuevo[13,14,15], Marc Suárez-Calvet[15,16,17], Estrella Morenas-Rodríguez[16,18], Gernot Kleinberger[16,18], Christian Haass[16,17,18], Michael Ewers[19], Johannes Levin[17,18,20], Martin R. Farlow[21,22], Richard J. Perrin[3,4,23,24], on behalf of the Alzheimer's Disease Neuroimaging Initiative (ADNI)¶, on behalf of the Dominantly Inherited Alzheimer Network (DIAN)¶, Carlos Cruchaga[1,2,23] *

1 Department of Psychiatry, Washington University School of Medicine, St. Louis, Missouri, United States of America, 2 Neurogenomics and Informatics Center, Washington University School of Medicine, St. Louis, Missouri, United States of America, 3 Department of Neurology, Washington University School of Medicine, St. Louis, Missouri, United States of America, 4 Knight Alzheimer Disease Research Center, Washington University School of Medicine, St. Louis, Missouri, United States of America, 5 Department of Psychiatry and Neurochemistry, Institute of Neuroscience and Physiology, The Sahlgrenska Academy at the University of Gothenburg, Mölndal, Sweden, 6 Clinical Neurochemistry Laboratory, Department of Neuroscience and Physiology, University of Gothenburg, Sahlgrenska University Hospital, Mölndal, Sweden, 7 Department of Neurodegenerative Disease, UCL Institute of Neurology, Queen Square, London, United Kingdom, 8 UK Dementia Research Institute at UCL, London, United Kingdom, 9 Department of Anesthesiology and Intensive Care Medicine, Sahlgrenska University Hospital, Mölndal, Sweden, 10 Institute of Clinical Sciences, The Sahlgrenska Academy at the University of Gothenburg, Gothenburg, Sweden, 11 Department of Internal Medicine, Institute of Medicine, The Sahlgrenska Academy at the University of Gothenburg, Göteborg, Sweden, 12 Neurology Department, Hospital de Sant Pau, Barcelona, Spain, 13 IDIBAPS, Alzheimer´s Disease and Other Cognitive Disorders Unit, Neurology Service, ICN Hospital Clinic, Barcelona, Spain, 14 Alzheimer´s Disease and Other Cognitive Disorders Unit, Neurology Service, ICN Hospital Clinic i Universitari, Barcelona, Spain, 15 BarcelonaBeta Brain Research Center, Pasqual Maragall Foundation, Barcelona, Spain, 16 Biomedical Center (BMC), Biochemistry, Ludwig-Maximilians-Universität München, Munich, Germany, 17 German Center for Neurodegenerative Diseases (DZNE), Munich, Germany, 18 Munich Cluster for Systems Neurology (SyNergy), Munich, Germany, 19 Institute for Stroke and Dementia Research, University Hospital, Ludwig-Maximilians-Universität München, Munich, Germany, 20 Department of Neurology, Ludwig-Maximilians-Universität München, Munich, Germany, 21 Indiana Alzheimer Disease Research Center, Indiana University School of Medicine, Indianapolis, Indiana, United States of America, 22 Department of Neurology, Indiana University School of Medicine, Indianapolis, Indiana, United States of America, 23 Hope Center for Neurological Disorders, Washington University School of Medicine, St. Louis, Missouri, United States of America, 24 Department of Pathology and Immunology, Washington University School of Medicine, St. Louis, Missouri, United States of America

¶ Membership of the Alzheimer's Disease Neuroimaging Initiative (ADNI) and Dominantly Inherited Alzheimer Network (DIAN) is provided in the Acknowledgments.
* cruchagc@psychiatry.wustl.edu



**Data Availability Statement:** All relevant data are within the paper and its Supporting information files.

## Abstract

Two genetic variants in strong linkage disequilibrium (rs9536314 and rs9527025) in the Klotho (*KL*) gene, encoding a transmembrane protein, implicated in longevity and

**Funding:** This work was supported by grants from the National Institutes of Health (R01AG044546, P01AG003991, RF1AG053303, R01AG058501, U01AG058922, RF1AG058501 and R01AG057777), the Alzheimer Association (NIRG-11-200110, BAND-14-338165, AARG-16-441560 and BFG-15-362540). This work was supported by access to equipment made possible by the Hope Center for Neurological Disorders, and the Departments of Neurology and Psychiatry at Washington University School of Medicine. The recruitment and clinical characterization of research participants at Washington University were supported by NIH P50 AG05681, P01 AG03991, and P01 AG026276. HZ is a Wallenberg Scholar supported by grants from the Swedish Research Council (#2018-02532), the European Research Council (#681712), Swedish State Support for Clinical Research (#ALFGBG-720931), the Alzheimer Drug Discovery Foundation (ADDF), USA (#201809-2016862), the AD Strategic Fund and the Alzheimer's Association (#ADSF-21-831376-C, #ADSF-21-831381-C and #ADSF-21-831377-C), the Olav Thon Foundation, the Erling-Persson Family Foundation, Stiftelsen för Gamla Tjänarinnor, Hjärnfonden, Sweden (#FO2019-0228), the European Union's Horizon 2020 research and innovation programme under the Marie Skłodowska-Curie grant agreement No 860197 (MIRIADE), and the UK Dementia Research Institute at UCL. KB is supported by the Swedish Research Council (#2017-00915), the Alzheimer Drug Discovery Foundation (ADDF), USA (#RDAPB-201809-2016615), the Swedish Alzheimer Foundation (#AF-742881), Hjärnfonden, Sweden (#FO2017-0243), the Swedish state under the agreement between the Swedish government and the County Councils, the ALF-agreement (#ALFGBG-715986), the European Union Joint Program for Neurodegenerative Disorders (JPND2019-466-236), the National Institute of Health (NIH), USA, (grant #1R01AG068398-01), and the Alzheimer's Association 2021 Zenith Award (ZEN-21-848495).

**Competing interests:** CC receives research support from: Biogen, EISAI, Alector and GSK. The funders of the study had no role in the collection, analysis, or interpretation of data; in the writing of the report; or in the decision to submit the paper for publication. CC is a member of the advisory board of Vivid genetics, Halia Therapeutics, Circular Genomics and ADx Healthcare. HZ has served at scientific advisory boards and/or as a consultant for Abbvie, Alector, Eisai, Denali, Roche, Wave, Samumed, Siemens Healthineers, Pinteon Therapeutics, Nervgen, AZTherapies, CogRx, and Red Abbey Labs, has given lectures in symposia

associated with brain resilience during normal aging, were recently shown to be associated with Alzheimer disease (AD) risk in cognitively normal participants who are *APOE* ε4 carriers. Specifically, the participants heterozygous for this variant (KL-SV$^{HET+}$) showed lower risk of developing AD. Furthermore, a neuroprotective effect of KL-VS$^{HET+}$ has been suggested against amyloid burden for cognitively normal participants, potentially mediated via the regulation of redox pathways. However, inconsistent associations and a smaller sample size of existing studies pose significant hurdles in drawing definitive conclusions. Here, we performed a well-powered association analysis between KL-VS$^{HET+}$ and five different AD endophenotypes; brain amyloidosis measured by positron emission tomography (PET) scans (n = 5,541) or cerebrospinal fluid Aβ42 levels (CSF; n = 5,093), as well as biomarkers associated with tau pathology: the CSF Tau (n = 5,127), phosphorylated Tau (pTau181; n = 4,778) and inflammation: CSF soluble triggering receptor expressed on myeloid cells 2 (sTREM2; n = 2,123) levels. Our results found nominally significant associations of KL-VS$^{HET+}$ status with biomarkers for brain amyloidosis (e.g., CSF Aβ positivity; odds ratio [OR] = 0.67 [95% CI, 0.55–0.78], β = 0.72, *p* = 0.007) and tau pathology (e.g., biomarker positivity for CSF Tau; OR = 0.39 [95% CI, 0.19–0.77], β = -0.94, *p* = 0.007, and pTau; OR = 0.50 [95% CI, 0.27–0.96], β = -0.68, *p* = 0.04) in cognitively normal participants, 60–80 years old, who are APOE e4-carriers. Our work supports previous findings, suggesting that the KL-VS$^{HET+}$ on an *APOE* ε4 genotype background may modulate Aβ and tau pathology, thereby lowering the intensity of neurodegeneration and incidence of cognitive decline in older controls susceptible to AD.

## Introduction

Alzheimer disease (AD), the most common form of dementia, affects about 30% of those aged over 85 years [1]. AD is classified as a neurodegenerative disease, affecting brain integrity and function, eventually resulting in progressive deterioration of cognitive capabilities [2]. Besides aging, a strong genetic risk factor for developing AD is the epsilon 4 allele of the apolipoprotein E gene (*APOE* ε4) [3, 4]. As such, participants carrying one or two *APOE* ε4 alleles are significantly overrepresented among persons diagnosed with AD, in comparison to non-carriers [5, 6]. This particular genetic variant has been shown to be associated with cognitive decline [7] and reduced mean age at onset even within families with late onset AD [8]. Further, *APOE* ε4 homozygosity among cognitively normal participants is associated with earlier and more abundant Aβ deposition [9–12], earlier pre-clinical memory decline [13], and an increased incidence of conversion to dementia [9], in comparison to *APOE* ε4 heterozygotes and non-carriers. These observations suggest that *APOE* ε4 influences the very core of AD pathophysiology, in large part due to its key role in fostering cerebral Aβ pathology. Therefore, identifying genetic factors that interact with *APOE* ε4 genotype to reduce Aβ burden and, eventually, a participant's risk for developing AD, may inspire novel strategies for preventing or halting the progression of AD and reveal novel targets for effective therapeutic interventions.

One such genetic factor that has been recently reported and showed a protective effect among cognitively normal *APOE* ε4 carriers is polymorphism on Klotho (*KL*) gene [6, 14, 15]. *KL* is a transmembrane protein that is cleaved by α- and β-secretases and shed into CSF and plasma, where it acts as a signaling molecule and longevity factor [16, 17] that promotes

sponsored by Cellectricon, Fujirebio, Alzecure and Biogen, and is a co-founder of Brain Biomarker Solutions in Gothenburg AB (BBS), which is a part of the GU Ventures Incubator Program. KB has served as a consultant, at advisory boards, or at data monitoring committees for Abcam, Axon, Biogen, JOMDD/Shimadzu. Julius Clinical, Lilly, MagQu, Novartis, Prothena, Roche Diagnostics, and Siemens Healthineers, and is a co-founder of Brain Biomarker Solutions in Gothenburg AB (BBS), which is a part of the GU Ventures Incubator Program, all unrelated to the work presented in this paper. JL reports speaker fees from Bayer Vital, Biogen and Roche, consulting fees from Axon Neuroscience and Biogen, author fees from Thieme medical publishers and W. Kohlhammer GmbH medical publishers, non-financial support from Abbvie and compensation for duty as part-time CMO from MODAG, outside the submitted work.

neuronal functions and brain resilience during aging [18–20]. In humans, two KL gene variants (rs9536314 for p.F352V and rs9527025 for p.C370S) exist in strong linkage disequilibrium and segregate together as a functional haplotype called KL-VS. Interestingly, heterozygosity for the KL-VS haplotype (KL-VS$^{HET+}$) has been associated with higher serum concentrations of KL [18, 21] which, in turn, has been reported to promote healthy brain aging and protect synaptic functions in comparison to participants who carry two copies of the KL-VS haplotype (KL-VS$^{HET-}$) [20, 22]. Even though KL-VS$^{HET+}$ is associated with better cognitive health and longevity among those aging normally, there exist no clear indication of its involvement in protection against aging-associated neurodegenerative disorders, such as AD.

Identification of genetic risk factors for AD based on clinical diagnosis poses several challenges. A clinical diagnosis of AD relies in part on evidence of cognitive decline using standard cognitive tests that might be influenced by factors unrelated to disease (e.g., anxiety, education, and general test-taking ability of the participant [23]). In addition, other salient factors that can play an important role include variability in the cognitive measures [24, 25], over-reliance on normative cut-off scores to diagnose dementia, and practice effect [24, 26]. A complementary approach to classical case-control studies involves the use of more robust and stable measures (endophenotypes) that support a diagnosis of AD, such as cerebrospinal fluid (CSF) biomarkers and Aβ burden assessed by positron emission tomography (PET). This approach can increase statistical power to identify AD genetic risk factors, discover novel associations, and understand their impact on the brain [27]. For example, by using brain endophenotypes, researchers have identified novel protective genetic variants in *TMEM106B* and *MS4A* genes, associated with neuroprotection (high neuronal proportion) in AD [28], and increased CSF soluble triggering receptor expressed on myeloid cells 2 (sTREM2) concentrations with reduced AD risk, respectively [29].

In the spectrum of AD pathology and different genetic factors that exert a protective effect in the context of disease onset and/or progression, KL-VS appears to be a compelling candidate due to its implication in promoting longevity and cognitive resilience during aging [18, 20, 22]. Interestingly, two recent studies evaluating the protective effect of KL-VS$^{HET+}$ against AD pathology in cognitively normal participants [15, 30] provided contradictory evidence. The first study [15] focused on 309 late-middle-aged adults (mean age 61 years) and found KL-VS$^{HET+}$ to be associated with reduced Aβ aggregation, suggesting its protective effect against *APOE* ε4-linked pathways to disease onset in AD. The second study [30] analyzed data from 581 adults (mean age 71 years) and found no significant associations between KL-VS$^{HET+}$ and cognitive decline, independent of the *APOE* ε4 genotype, suggesting no modifying effect of KL-VS$^{HET+}$ on Aβ aggregation and *APOE* ε4-driven cognitive decline in preclinical AD. Furthermore, a study assessing the association between Klotho KL-VS haplotype and cognition using data from the population-based Heinz Nixdorf Recall Study in 1812 subjects (55–87 years) suggested a slightly lower cognitive performance in KL-VS$^{HET+}$ subjects [31]. However, a recent large-scale meta-analysis [6] that focused on cognitively normal participants in the age range of 60–80 years revealed a 30% reduction in AD risk for participants who are *APOE* ε4 carriers and KL-VS$^{HET+}$. They also observed a significant associations between KL-VS$^{HET+}$ and higher Aβ42 in CSF ($p = 0.03$) and between KL-VS$^{HET+}$ and lower Aβ on PET scans ($p = 0.04$), modulated by *APOE* ε4 status. Focusing on tau pathology, a recent study showed KL-VS$^{HET+}$ to be associated with both lower levels and slower rate of change in amyloid-related increase of tau-PET accumulation in asymptomatic and symptomatic elderly participants, supporting a protective effect of KL-VS$^{HET+}$ on the primary AD pathologies [32]. Due to contradictory outcomes from the existing reports and their relatively small sample sizes, we performed a systematic evaluation of association between KL-VS$^{HET+}$ and multiple well-established AD endophenotypes to evaluate whether it has a protective effect on AD pathology in

asymptomatic elderly participants. Here, we have performed a well-powered association analysis between KL-VS$^{HET+}$ and five different AD endophenotypes: Aβ assessed by PET scans (n = 5,541) and CSF (n = 5,093), as well as the CSF Tau (n = 5,127), phosphorylated Tau (pTau181; n = 4,778) and sTREM2 (n = 2,123). In line with previous studies, we performed *APOE* ε4- and age-stratified (60–80 years) analyses to determine if there is any association between KL-VS$^{HET+}$ and Aβ aggregation that is modulated by *APOE* ε4 status. In addition, we also evaluated if there is any association between KL-VS$^{HET+}$ and other AD endophenotypes that include Tau, pTau, and sTREM2 measured by CSF. Briefly, in the case of *APOE* ε4-carriers, we found significant associations between KL-VS$^{HET+}$ and biomarkers for brain amyloidosis (CSF Aβ42; *p* = 0.007) and tau pathology (CSF Tau; *p* = 0.007, and pTau; *p* = 0.04). As evident from the observed *P*-values, the detected associations are nominally significant and would likely fail multiple test correction, indicating the need for validating these findings in studies with even larger sample sizes before drawing definitive conclusions. Nevertheless, these findings suggest that KL-VS$^{HET+}$ exerts an *APOE* ε4-genotype dependent protective effect on CSF Tau and pTau concentrations and on subsequent cognitive decline in older cognitively normal participants susceptible to AD.

## Methods

### Study samples and phenotype processing

Written consent was obtained for all participants. This study was approved the Washington University Human study committee. IRB approval #: 201109148.

For this study, we collected data from 17 different AD-related cohorts. Participants were enrolled in the Memory and Aging Project (MAP) at the Knight Alzheimer's Disease Research Center (Knight-ADRC), Alzheimer's Disease Neuroimaging Initiative (ADNI, adni.loni.usc. edu), BIOCARD, the Dominantly Inherited Alzheimer Network (DIAN), HB, Hospital Sant Pau (Lleo), London, MOLI, Pau, Mayo Clinic (Mayo), SWEDEN, UPENN, UW, Parkinson's Progression Markers Initiative (PPMI), Anti-Amyloid Treatment in Asymptomatic Alzheimer's Disease (A4), and ADNI Department of Defense (ADNIDOD) studies. Total sample size was 9,526 (S1 Table in S2 File). Collection of genotype data, PET image processing, and CSF data processing for each cohort are described in detail in the respective studies [10, 11, 28, 29, 33, 34]. We analyzed the association between KL-VS$^{HET+}$ and five different AD endophenotypes (Table 1) that served as biomarkers for brain amyloidosis (Aβ pathology assessed by amyloid-PET [n = 5,541] and Aβ42 measured from CSF [n = 5,093]), tau pathology (Tau [n = 5,127] and pTau181 [n = 4,778] from CSF), and inflammation (sTREM2 levels from CSF [n = 2,123]). A schematic overview of the analyses conducted and the datasets used is provided in Fig 1.

Briefly, participants were diagnosed as cognitively normal (controls) or AD (cases), based on the clinical dementia rating (CDR) that was available for 88% of the total dataset. The CDR is a five-point scaling system that describes the overall dementia severity for each participant (no dementia = 0, very mild = 0.5, mild = 1, moderate = 2, and severe = 3). Participants with CDR = 0 were categorized as controls, and those with CDR > 0 were defined as cases. Any participant who was missing information about age, sex, KL-VS$^{HET+}$, *APOE* ε4 genotype, or genetic principal components (PCs) was excluded from the analysis. Following this rationale, we considered 3,725 cognitively normal participants assessed by amyloid-PET and 1,030 cognitive normal participants measured by CSF (Aβ42, Tau, and pTau) and 639 participants with CSF sTREM2 levels. Similarly, the number of clinically defined AD participants assessed from amyloid-PET, CSF Aβ42, Tau, pTau, and sTREM2 were 1,090, 2,424, 2,443, 2,297, and 1,074, respectively.

**Table 1. Demographics of analyzed Alzheimer's disease (AD) endophenotypes.**

|  | Total | Amyloid-PET | Aβ42 | Tau | pTau181 | sTREM2 |
|---|---|---|---|---|---|---|
| **Sample size** | 9,526 | 5,541 | 5,093 | 5,127 | 4,778 | 2,123 |
| **Female (%)** | 51.86 | 54.54 | 49.30 | 49.19 | 48.68 | 50.31 |
| **Age, mean (SD)** | 68.93 (11.13) | 69.53 (10.73) | 67.04 (13.27) | 67.15 (13.30) | 66.93 (13.43) | 68.17 (12.13) |
| *APOE* ε4+ (%) | 39.03 | 37.39 | 40.74 | 41.08 | 39.47 | 42.11 |
| **Biomarker, mean (SD)** | 0.028 (0.02) | 0.045 (1.03) | -0.0018 (1) | 0.027 (1) | 0.02 (1) | 0.05 (0.98) |
| **Klotho-VS[HET+] (%)** | 25.76 | 25.99 | 25.31 | 25.16 | 25.09 | 25.20 |
| **Cases** | 3,109 | 1,090 | 2,424 | 2,443 | 2,297 | 1,074 |
| **Controls** | 5,286 | 4,117 | 1,584 | 1,589 | 1,582 | 879 |

Demographics of participants at the time of amyloid PET imaging and CSF sampling. This table summarizes basic demographic information of participants included in the analysis. For each modality, we report percentage of females, mean age of the participants and standard deviation (SD) in the age, percentage of *APOE* ε4-carriers (*APOE* ε4+) participants, mean value of the endophenotypic biomarker and its SD, percentage of KL-VS heterozygous (KL-VS[HET+]) participants, and number of cases and controls. Samples with missing case/controls status were also considered in the 'all participants' analysis. To normalize endophenotypes across different cohorts, we converted different amyloid imaging measures (e.g., Centiloid, PiB, and AV45) into log-normalized z-score using "scale" function in base R. Phenotype from each cohort was normalized individually to account for within cohort variation. These AD endophenotypes are used for checking their association with KL-VS[HET+]. Abbreviations: PET, positron emission tomography; Aβ, β-amyloid; pTau, phosphorylated tau; soluble triggering receptor expressed on myeloid cells 2, sTREM2; sd, standard deviation; KL-VS, Klotho-VS; Het+, heterozygosity.

For each cohort, amyloid PET images were normalized to their reference cerebellar regions to obtain standardized uptake value ratios (SUVR) in a composite of cortical brain areas. The normalized z-scores were calculated for each endophenotype using the mean and standard deviation (SD) units across each cohort and applied to the entire endophenotype. These normalized z-scores were used for dichotomizing each endophenotype into biomarker positive

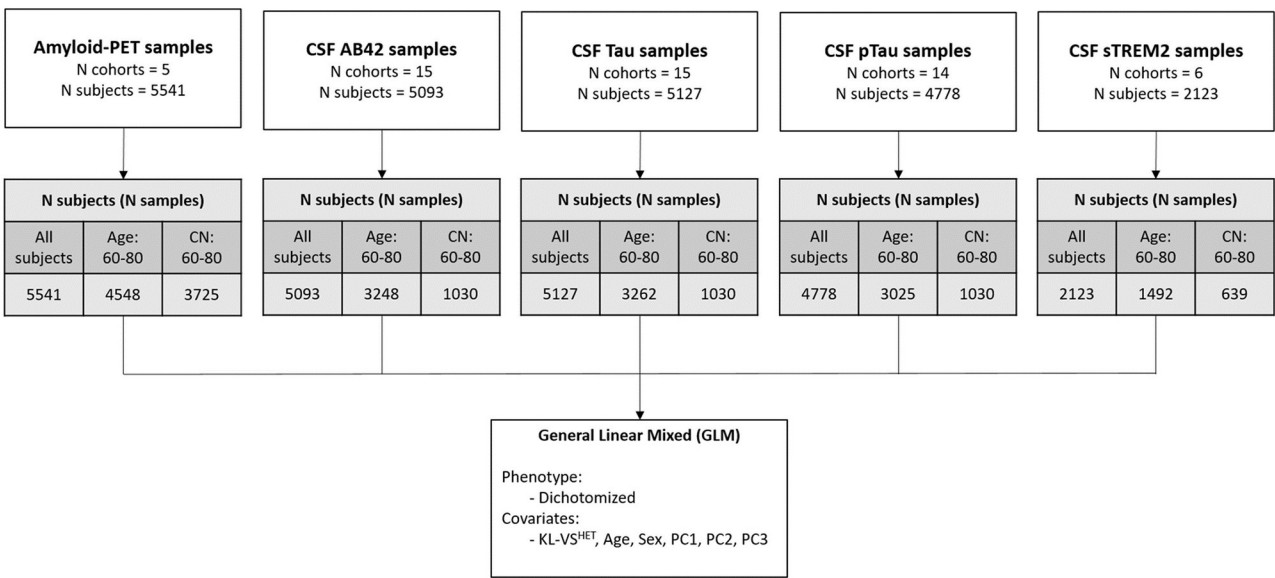

**Fig 1. Schematic overview of datasets and performed analysis.** Number of participants in each modality were stratified into three categories: 1) All of the participants; 2) Age: 60–80, participants aged 60 to 80 years; 3) CN: 60–80, cognitively normal participants aged 60 to 80 years. Association between KL-VS[HET] and endophenotypes were assessed using generalized linear mixed (logistic regression) model for dichotomized phenotype. Age, sex, and first three genetic PCs were used as covariates in an *APOE* ε4-stratified analysis. Abbreviations: PET, positron emission tomography; N, number of; CSF, cerebrospinal fluid; Aβ, β-amyloid; pTau, phosphorylated tau181; soluble triggering receptor expressed on myeloid cells 2, sTREM2; CN, cognitively normal; KL, Klotho; Het, heterozygous; PC, principal component.

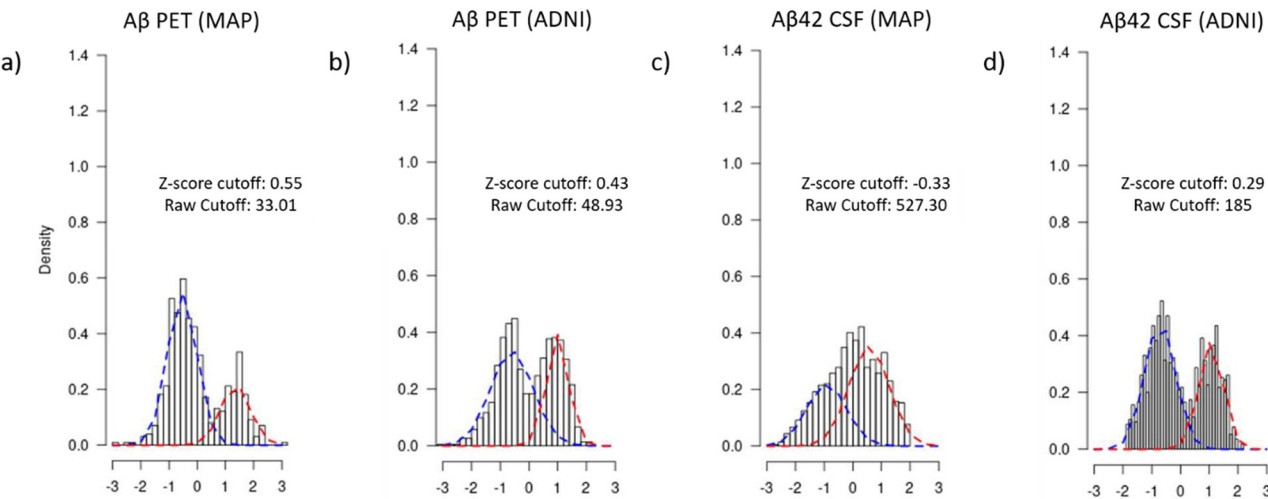

**Fig 2. Cutoffs for dichotomizing different AD endophenotypes across MAP and ADNI cohorts.** A density plot defining the dichotomization cutoffs for Aβ assessed by PET scan and Aβ42 from CSF in MAP and ADNI cohorts. The distribution of z-score for cases and controls is shown by red and blue dotted lines, respectively. The cut-off point where both these distributions overlap was selected as the dichotomization threshold for each endophenotype. The dichotomization was performed on normalized z-scores for each endophenotype. However, the corresponding raw score for each dichotomization threshold is also labelled in the plot. Abbreviations: Aβ, β-amyloid; PET, positron emission tomography; MAP, Memory and Aging Project; ADNI, Alzheimer's Disease Neuroimaging Initiative; CSF; Cerebrospinal fluid.

(case) and negative (control), as previously described [11]. Briefly, defining biomarker positivity and negativity requires the selection of a cut-point. We and others [11, 35, 36] have demonstrated that it is possible to use Gaussian mixture model (GMM) to statistically infer that cutoff. We overlapped the distributions of quantitative z-scores from cases and controls for each endophenotype and employed a GMM that relies on hierarchical model-based agglomerative clustering to get votes for defining a cut-point for dichotomization. We used Mclust function from "mclust" R package (version 5.4.6) for dichotomizing all quantitative endophenotypes separately. The empirical dichotomization cutoffs obtained using this approach appeared consistent with existing literature. For instance, our model suggested the cut-point of 527 pg/mL for Aβ42 from CSF in MAP cohort which is between 500 pg/mL [37] to 518 ng/l [35] depending on the study. A density plot defining the dichotomization cutoffs for Aβ assessed by PET scan and Aβ42 from CSF in MAP and ADNI cohorts is shown in Fig 2. Further details about the empirical cutoffs derived from z-scores and their corresponding raw values are provided in S2 Table in S2 File. The dichotomized (biomarker positive/negative) endophenotypic status was used as a response variable to assess its association with Klotho heterozygosity.

## Genotyping, quality checks, imputation, and population structure

We applied stringent quality control (QC) steps to process the genotyping array and sequencing data. We used the threshold of 98% for removing single nucleotide polymorphisms (SNPs) and participants with low call rate. Autosomal SNPs that were not in the Hardy-Weinberg equilibrium ($P < 1 \times 10^{-6}$) were also removed. Duplication and relatedness of participants were estimated from identity-by-descent (IBD) analysis carried out in Plink version 1.9 [38]. In case of related participants (Pihat ≥0.25), the samples from MAP or with a higher number of variants that passed the QC were prioritized. For phasing and imputation, we used The 1000 Genomes Project Phase 3 data (October 2014), SHAPEIT v2.r837 [39], and IMPUTE2 v2.3.2 [40]. We used imputed probability score < 0.90 and ≥0.90 as thresholds for missing and fully observed participant genotypes, respectively. Genotyped and imputed variants with

MAF < 0.02 or IMPUTE2 information score < 0.30 were discarded. Principle component analysis (PCA) was performed on the genotype data to obtain genetic PCs that capture population substructure (S2 Fig in S1 File). To obtain the largest and most homogeneous pool of population, only European participants were considered (S3 Fig in S1 File) for the subsequent statistical analyses.

## Statistical analyses

Statistical analyses and data visualization were performed in Plink version 1.9 [38] and R version 3.5.2 [41]. We performed association analyses of KL-VS$^{HET+}$ status with different AD endophenotypes from PET scan (Aβ) and CSF (Aβ42, Tau, pTau, and TREM2). The associations between the biomarker positivity and KL-VS$^{HET+}$ were tested using logistic regression model. The implementation of regression model from the base R [41] "stats" package was used for the evaluation of association and the outcomes measurements were adjusted for sex, age, and first three genetic PCs. For the Aβ levels measured by PET scan, we considered the age at scan and in the case CSF biomarker levels, age at lumbar puncture. Furthermore, associations were evaluated across three different strata: (1) all participants (AD and controls); (2) those aged 60 to 80 years (AD and controls); and (3) only cognitively normal participants aged 60 to 80 years. All association analyses were stratified by *APOE* ε4 status: *APOE* ε4 carriers (APOE-24, 34, and 44) and *APOE* ε4 non-carriers (APOE- 22, 32, and 33). Associations were deemed significant at a threshold of $p < 0.05$.

## Results

The primary aim of this study was to determine whether there is a significant association between KL-VS$^{HET+}$ and AD endophenotypes in cognitively normal participants who are *APOE* ε4-carriers. In addition, we also assessed the KL-VS effects in participants with AD dementia. To accomplish this goal, we analyzed genetic data and five different AD endophenotypes from 9,526 participants obtained through 17 different AD-related cohorts. In line with existing studies [6, 15], our findings validate the protective effect of KL-VS$^{HET+}$ against AD in cognitively normal participants who are *APOE* ε4 carriers.

## Association between KL-VS$^{HET+}$ status and brain amyloidosis measured by PET scan and CSF

We evaluated the association of KL-VS$^{HET+}$ status with Aβ pathology as measured by amyloid-PET and CSF Aβ42 for 5,541 and 5,093 participants, respectively (Table 1). Although associations were evaluated for three different age ranges (S3 Table in S2 File), our main focus was cognitively normal participants who are 60 to 80 years old (Table 2). We focused on this age range to be consistent with existing studies that report a pronounced effect of *APOE* ε4 positivity on AD dementia risk between age 60 to 80 years in comparison to older participants (≥80 years) [6, 42]. For Aβ levels measured by PET, we observed that KL-VS$^{HET+}$ status was more prevalent in Aβ biomarker negative participants in comparison to the positive ones, but there was no significant association within any age group or *APOE* ε4 strata (S3 Table in S2 File). On the other hand, we found a significant association between KL-VS$^{HET+}$ status and CSF Aβ biomarker positivity in cognitively normal *APOE* ε4-carrier participants who are 60 to 80 years old (Table 2; OR = 0.67 [95% CI, 0.55–0.78], β = 0.72, $p = 0.007$). We also observed a significant association for *APOE* ε4 non-carriers; however, the effect size and the strength of the association was lower in this group (OR = 0.61 [95% CI, 0.51–0.70], β = 0.46, $p = 0.03$) than the *APOE* ε4-carriers (Table 2). Taken together, we were able to replicate the previously

**Table 2. Genetic association of KL-VS$^{HET+}$ with AD endophenotypes in cognitively normal participants aged 60–80 years, stratified by *APOE* ɛ4 status.**

| Modality | Group | CN participants (KL-VS$^{HET+}$ %) | Odds ratio | Estimate | P value |
|---|---|---|---|---|---|
| **Amyloid PET** | APOE4+ | 1328 (27.5) | 0.94 | -0.07 | 0.61 |
| | APOE4- | 2397 (26.5) | 0.99 | -0.01 | 0.90 |
| **Aβ42** | APOE4+ | 308 (26.3) | 0.67 | 0.72 | **0.007** |
| | APOE4- | 722 (26.3) | 0.61 | 0.46 | **0.03** |
| **Tau** | APOE4+ | 308 (26.9) | 0.39 | -0.94 | **0.007** |
| | APOE4- | 722 (26.5) | 0.85 | -0.16 | 0.49 |
| **pTau** | APOE4+ | 308 (26.9) | 0.50 | -0.68 | **0.04** |
| | APOE4- | 722 (26.3) | 0.89 | -0.11 | 0.61 |
| **sTREM2** | APOE4+ | 199 (31.2) | 1.08 | 0.08 | 0.80 |
| | APOE4- | 440 (25.2) | 1.20 | 0.18 | 0.43 |

Association between KL-VS$^{HET+}$ and different dichotomized AD endophenotypes were assessed using logistic regression model. We used dichotomized endophenotype as the response variable, whereas, age, sex, and first three genetic PCs were used as covariates in an *APOE*4-stratified analysis. Significant associations are represented by bold P-values. Abbreviations: KL-VS$^{HET+}$, Klotho-VS heterozygous; CN, cognitively normal; AD, Alzheimer's disease; Std. Error, Standard error; %, percentage; Aβ, β-amyloid; pTau, phosphorylated tau181; soluble triggering receptor expressed on myeloid cells 2, sTREM2.

reported associative findings [6, 42] between increased Aβ42 CSF levels and KL-VS$^{HET+}$ in a larger sample group (S1 Fig in S1 File).

## *APOE* ɛ4-related alteration in tau pathology varies by KL-VS$^{HET+}$ status

Similar to Aβ association analyses, we also determined if there was an age- and *APOE* ɛ4-dependent association of KL-VS$^{HET+}$ status with dichotomized CSF Tau and pTau biomarker positivity. In the age range of 60 to 80 years, we observed significant association between KL-VS$^{HET+}$ status and CSF Tau (OR = 0.39 [95% CI, 0.20–0.77], *p* = 0.007), and pTau (OR = 0.50 [95% CI, 0.27–0.96], *p* = 0.04) levels in elderly (60–80 years) cognitively normal *APOE* ɛ4-carrier participants (Table 2). In both cases, the KL-VS$^{HET+}$ was associated with biomarker negative status (e.g., lower CSF Tau or pTau levels which are associated with lower AD risk (β = -0.94 and -0.68 for Tau and pTau, respectively). Regarding the effect size, we observed an almost 6-fold decrease for both Tau and pTau levels in *APOE* ɛ4-carriers as compared to the non-carriers, suggestive of a more pronounced protective effect for participants carrying one copy of KL-VS haplotype. As expected, a consistent negative association was observed for both modalities when effect sizes were represented in the form of a Forest plot (Fig 3). While we observed similar trends in the effect of KL-VS$^{HET+}$ on tau pathology biomarker negativity for cognitively normal participants who are *APOE* ɛ4 non-carriers (β = -0.16 and -0.11 for Tau and pTau, respectively), the association was not deemed significant (*p* = 0.49 and 0.61 for Tau and pTau, respectively).

To further elucidate the link between *APOE* ɛ4 and KL-VS$^{HET+}$, we performed the association analysis including an interaction effect for the presence or absence of *APOE* ɛ4 and the status of each biomarker (positive or negative). Results did not showed any significant association across any endophenotype (S4 Table in S2 File), emphasizing the *APOE* ɛ4-dependent association of KL-VS$^{HET+}$ and different AD endophenotypes. In addition, we also assessed associations between *APOE* ɛ4 and AD pathologies in the group with 0 and 2 copies of KL-VS haplotype (KL-VS$^{HET-}$). We observed no significant association across any AD endophenotypes (S5 Table in S2 File), confirming the existing hypothesis that suggests the neuroprotective effect of KL heterozygosity (KL-VS$^{HET+}$) but not the homozygosity (KL-VS$^{HET-}$) [20, 22].

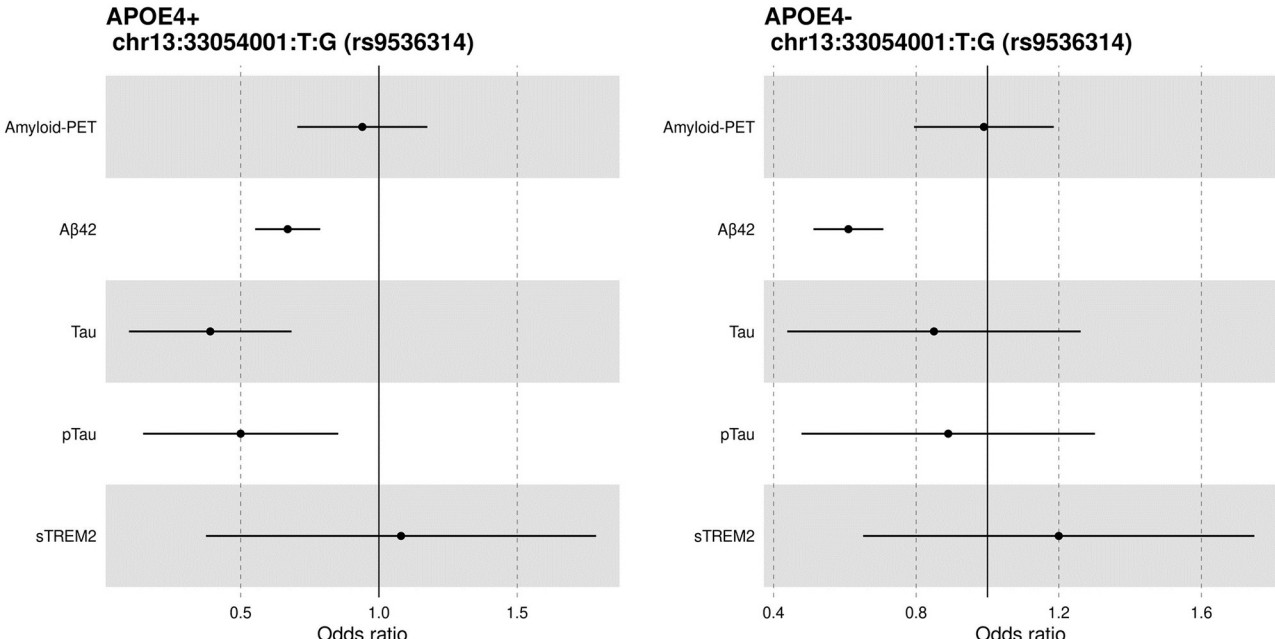

**Fig 3. Forest plot of odds ratio (OR) for KL-VS^HET+ association with dichotomized AD endophenotypes in 60–80 year cognitively normal participants, stratified by *APOE* ε4 status.** A significant association was detected between KL-VS^HET+ and dichotomized Aβ, Tau, and pTau CSF levels. In case of Aβ, the associations were deemed significant across both *APOE* ε4 strata, whereas for Tau and pTau, associations were observed only in *APOE* ε4-carriers, representing an exclusive protective effect of KL-VS^HET+ for the cognitively normal participants aged 60 to 80 years and carrying *APOE* ε4. Abbreviations: *APOE*4+, *Apolipoprotein E*4 positive; *APOE*4-, *Apolipoprotein E*4 negative; PET, positron emission tomography; Aβ, β-amyloid; pTau, phosphorylated tau181; soluble triggering receptor expressed on myeloid cells 2, sTREM2.

## Direct effect of KL-VS^HET+ status on inflammation-specific biomarker

For sTREM2 CSF levels, we did not observe any significant association between this inflammation biomarker and KL-VS^HET+ status across any participant stratification, regardless of the age group and *APOE* ε4 status. Nevertheless, for cognitively normal participants aged 60 to 80 years, we observed that KL-VS^HET+ status is associated with increased sTREM2 CSF levels, which represents the positivity of inflammation-specific biomarker, but association was not deemed significant in *APOE* ε4-carriers (OR = 1.08 [95% CI, 0.58–2], β = 0.08, *p* = 0.80) as well as non-carriers (OR = 1.20 [95% CI, 0.77–1.86], β = 0.18, *p* = 0.43).

## Sensitivity analysis result: Robustness of the associations between KL-VS^HET+ status and brain amyloidosis to the *APOE* ε4 status, age, and sex

In order to estimate whether the associations between KL-VS^HET+ and amyloidosis were confounded by uneven sample size of *APOE* ε4-carriers and non-carriers as well as by differences in the sex and age of the participant groups, the association analyses were repeated with equal numbers of *APOE* ε4-carriers (N = 308) and non-carriers (N = 308) matched for age and sex. As in the full-sample analyses for cognitively normal participants aged 60 to 80, these smaller, balanced analyses revealed that KL-VS^HET+ was consistently associated with CSF Aβ biomarker positivity among *APOE* ε4-carriers (OR = 0.68 [95% CI, 0.56–0.78], β = 0.75, *p* = 0.005; S6 Table in S2 File) and among *APOE* ε4 non-carriers (OR = 0.69 [95% CI, 0.51–0.79], β = 0.70, *p* = 0.034). Likewise, similar trends were observed between the full-sample and smaller, balanced analyses for amyloid imaging, CSF Tau, pTau181, and sTREM2 (S6 Table in

S2 File). In these corrections for class imbalance of *APOE* ε4-carriers and non-carriers, as well as for males and females of same age, the direction of effect remains the same for each endophenotype, and the strength of association becomes more profound (lower p-values). Taken together, these results suggest that observed associations between KL heterozygous cognitively normal participants and different AD endophenotypes (e.g. tau and pTau) are independent of the distribution of *APOE* ε4 carriage status, age, and sex of participants.

## Cognitively normal participants (aged 60 to 80) drive association between KL-VS[HET +] status and AD endophenotypes

We also conducted *APOE* ε4-stratified association analyses between KL-VS[HET+] status and biomarkers for brain amyloidosis (Aβ from PET and CSF), tau-related pathology (CSF Tau and pTau), and inflammation (CSF sTREM2) for all participants, regardless of their age and case-control status. In these larger inclusive analyses, no significant association was observed between KL-VS[HET+] and any of the five endophenotypes (S3 Table in S2 File). Similarly, we also performed the same analyses but restricting to participants between 60 and 80 years of age, regardless of their case-control status; even in that case, no significant association was detected across any endophenotype (S3 Table in S2 File). Only when participants were restricted to the age range 60–80 and cognitive normalcy were significant association detected between KL-VS[HET+] status and CSF Aβ, Tau, and pTau levels (S3 Table in S2 File). Notably, these findings suggest that the nearly-significant associations observed in the more inclusive analyses (e.g. for Aβ42) were mainly driven by the cognitively normal participants who are 60 to 80 years old.

## Discussion

The role of Klotho protein as a longevity factor is widely recognized [16, 17]. There has been an increasing amount of evidence supporting the relationship between KL-VS[HET+] and preserved brain integrity and cognitive performance during normal aging [18–20]. In this study, we examined the association of KL-VS[HET+] status with five different AD-related endophenotypes that serve as biomarkers for brain amyloidosis (Aβ levels measured from CSF and amyloid PET), tau pathology (CSF Tau and pTau), and inflammation (sTREM2). To our knowledge, we have analyzed the largest sample size of AD endophenotypic data for evaluating its association with KL-VS[HET+]; this approach is instrumental to discern the potential protective effect of this heterozygous genetic variant for AD in cognitively normal *APOE* ε4-carriers. Our results showed that KL-VS[HET+] status was associated with CSF Aβ42, Tau and pTau biomarker negativity in participants who are cognitively normal *APOE* ε4-carriers within an age range of 60 to 80 years. This finding suggests that KL-VS[HET+] status reduces the risk of subsequent AD dementia among *APOE* ε4-carriers by lowering the AD pathology burden [6, 42].

We were able to replicate the findings by Belloy et al. [6]; that is, KL-VS[HET+] status was significantly associated with increased CSF Aβ42 levels (Aβ biomarker positivity) for cognitively normal participants aged 60 to 80 years who are *APOE* ε4-carriers (OR = 0.67 [95% CI, 0.55–0.78], β = 0.72, $p$ = 0.007). Further, our analyses also found this association to be significant among *APOE* ε4 non-carriers (OR = 0.61 [95% CI, 0.51–0.70], β = 0.46, $p$ = 0.03), with similar (overlapping 95% CI) effect sizes (Table 2). Although no significant association was observed with amyloid PET, the detected trend towards a negative association suggests that KL-VS[HET+] may protect against AD by reducing deposits of Aβ that are capable of binding amyloid PET tracers. Consistently, studies have shown a very high concordance between CSF Aβ42 and amyloid PET [43], but with a proportion of individuals with discordant results (CSF+/PET-); such individuals may represent the earliest stages of AD neuropathologic change, when low CSF Aβ levels appear to coincide with early amyloid deposition, but amyloid deposits have not

yet accrued sufficiently to reach threshold for amyloid-PET tracer binding, and neurodegeneration has not yet begun [44, 45]. In support of this interpretation, a previous study investigating longitudinal differences in cognition between participants without dementia with different CSF and PET profiles found no memory decline in concordant-negative (CSF−/PET−) and discordant (CSF+/PET−) groups, in contrast to the concordant-positive (CSF+/PET+) group that deteriorated over time [46]. Furthermore, Palmqvist et al. [44] reported similar results, when they analyzed 437 non-demented participants from ADNI whose results from amyloid PET scans and CSF Aβ measurements showed that CSF Aβ levels become abnormal in the earliest stages of AD, before amyloid PET and before neurodegeneration starts.

We also investigated whether KL-VS$^{HET+}$ status is significantly associated with Tau and pTau levels in CSF. We found that KL-VS$^{HET+}$ status was significantly associated with decreased levels of CSF Tau (OR = 0.39 [95% CI, 0.20–0.77], β = -0.94, $p$ = 0.007) and pTau (OR = 0.50 [95% CI, 0.27–0.96], β = -0.68, $p$ = 0.04) i.e., CSF Tau and pTau biomarker negativity, in participants who are *APOE* ε4-carriers and 60 to 80 years old. Interestingly, *APOE* ε4 non-carriers showed similar negative trends, but the associations were not significant for Tau (OR = 0.85 [95% CI, 0.54–1.35], β = -0.16, $p$ = 0.49) or pTau (OR = 0.89 [95% CI, 0.57–1.39], β = -0.11, $p$ = 0.61). This indicates that KL-VS$^{HET+}$ status interaction with pathological aspects of AD are more profound among *APOE* ε4-carriers, such as Aβ and Tau accumulation during the pre-clinical phase of the disease [47, 48]. Although we did not find a protective effect of KL-VS$^{HET+}$ in individuals with AD dementia, a recent study reported that KL-VS$^{HET+}$ attenuated the association between higher amyloid PET and higher increases in tau PET accumulation [32]. Reasons for the discrepancy could be that here we did not investigate interaction effects of KL-VS$^{HET+}$ and amyloid on tau pathology, and second, we did not investigate tau PET, which assesses fibrillar tau deposits, whereas CSF p-tau181 appears to represent one or more earlier phenomena that do not closely correlate with neurofibrillary tangle burden. Indeed, previous studies have also reported a significant association between KL-VS heterozygosity and reduced tau accumulation and lower memory impairment in elderly humans at risk of AD dementia [32, 49]. However, in a mouse model of AD that was used to examine the neuroprotective effects of Klotho protein against neuronal damage associated with oxidative stress and neurodegeneration, no changes in Tau phosphorylation were observed in the presence of Klotho [50]. Unlike Tau and pTau association with KL-VS$^{HET+}$ status, we observed a positive association with CSF levels of sTREM2 (inflammation biomarker). The observed increase in CSF sTREM2 levels was not significantly associated with KL-VS$^{HET+}$ status for either *APOE* ε4-carriers (OR = 1.08 [95% CI, 0.59–2], β = 0.08, $p$ = 0.80) or non-carriers (OR = 1.20 [95% CI, 0.77–1.86], β = 0.18, $p$ = 0.43). Interestingly, recent studies [29, 34, 51, 52] have shown that higher sTREM2 levels are associated with lower AD risk and slower progression. Therefore, the observed positive association suggests that the protective effect of the KL-VS$^{HET+}$ might be mediated by higher CSF sTREM2 levels. However, this hypothesis will need to be validated in studies with larger sample size for CSF sTREM2 levels.

Taken together, the observed significant associations between KL-VS$^{HET+}$ status and biomarkers for brain amyloidosis (CSF Aβ positivity) and tau pathology (CSF Tau and pTau negativity) are suggestive of neuroprotective effect of KL-VS$^{HET+}$ against age-related biomarker, biomolecular, and cognitive alterations that confer risk for AD. Our results further strengthen the findings of a recent meta-analysis including 25 independent studies, showing that *APOE* ε4-carriers who were also KL-VS$^{HET+}$, were at a reduced risk for the combined outcome of conversion to mild cognitive impairment (MCI) or AD [6]. Besides, several other studies that evaluated the association of KL-VS$^{HET+}$ status with different cognitive measures in control participants did not consider interactions with *APOE* ε4 but did observe protective associations that were more pronounced closer to 80 years of age [18, 53, 54].

Notably, we assessed the associations between KL-VS[HET+] status and AD endophenotypes across three age strata: all of the participants (AD and controls); only those aged 60 to 80 years (AD and controls); and only cognitively normal participants aged 60 to 80 years (S3 Table in S2 File). Owing to a higher genetic risk for AD attributable to *APOE* ε4 in individuals who are 60 to 80 years old [55–57] and an existing study that hypothesized protective association of KL-VS[HET+] status to be strongest in *APOE* ε4 carriers who are 60 to 80 years old [6], the a priori focus of the current study was also at this particular age range. Although we observed similar associative trends most of the time, it was interesting to see how the effects became apparent when restricting analyses to the cognitively normal participants in the age range of 60 to 80 years. In all of the cases, no significant associations were observed between KL-VS[HET+] status and AD endophenotypes while considering all of the AD and cognitively normal participants, or all of the AD and cognitively normal participants within age range of 60–80. However, pronounced effects and associations were apparent for Aβ, Tau, and pTau levels from CSF, while considering cognitively normal participants who are *APOE* ε4-carriers and KL-VS[HET+]. These findings suggest that the cognitively normal participants group, aged 60 to 80 years, mainly drove the outcome in our analyses, further strengthening the existing hypothesis that KL-VS heterozygous genotype is favorable for better health and cognition in older people [18, 53, 58]. Importantly, we also observed that associations between KL heterozygous cognitively normal participants and different AD endophenotypes are robust to uneven sample size of *APOE* ε4-carriers and non-carriers as well as differences in the sex and age of the participants (S6 Table in S2 File). Although, the observed findings appear consistent with our initial hypothesis and confirm existing literature [6, 42], the detected associations are nominally significant and would likely fail multiple test correction due to limited sample size. Therefore, additional studies are required to investigate the associations between KL-VS[HET+] and AD endophenotypes with relatively larger sample size to draw definitive conclusions.

The exact mechanism underlying the KL-VS[HET+] interaction with *APOE* ε4 and modulation of Aβ, Tau, and pTau burden is yet unknown. However, it is logical to postulate that KL-VS[HET+] may confer resilience by increasing the serum level of circulating Klotho protein [18, 21] or by changing its function. In animal mouse models, elevated klotho levels have led to an extended lifespan [17], enhanced cognition [19] and increased resilience to AD-related toxicity [58]. Other studies in humans indicated that KL-VS[HET+] status has protective effects against brain aging and cognitive decline [21, 59], suggestive of its protective association against AD. Our findings also suggest that middle-aged *APOE* ε4-carriers who are KL-VS[HET+] might show resilience to age-induced cognitive and tau changes. Interestingly, we have observed an age-specific association between KL-VS[HET+] and AD endophenotypes, which is in line with existing studies reporting a specific time window for the effect of KL-VS polymorphism [20, 59].

To conclude, our work contributes to the existing literature by demonstrating that the protective effects of KL-VS[HET+] extend to AD-related Aβ, Tau, and pTau endophenotypes and deficits in memory and executive function in cognitively normal *APOE* ε4-carriers who are at risk for developing AD. One promising research avenue for the future studies could be to assess whether Klotho protein levels in the CSF or serum/plasma of participants associate with measures of preclinical and symptomatic AD.

## Supporting information

**S1 File. File containing all supplementary figures for the study.**
(DOCX)

**S2 File. File containing all supplementary tables for the study.**
(DOCX)

## Acknowledgments

We thank all the participants and their families, as well as the many involved institutions and their staff. Data used in preparation of this article were also obtained from the Dominantly Inherited Alzheimer Network (DIAN) and Alzheimer's Disease Neuroimaging Initiative (ADNI) consortiums (adni.loni.usc.edu). As such, the investigators within the ADNI contributed to the design and implementation of ADNI and/or provide data but did not participate in analysis or writing of this report. A complete listing of ADNI investigators can be found at: http://adni.loni.usc.edu/wp-content/uploads/how_to_apply/ADNI_Acknowledgement_List. pdf.

## Author Contributions

**Conceptualization:** Muhammad Ali, Carlos Cruchaga.

**Data curation:** Yun Ju Sung, Fengxian Wang, Maria V. Fernández.

**Formal analysis:** Muhammad Ali.

**Funding acquisition:** John C. Morris, Anne M. Fagan, Carlos Cruchaga.

**Project administration:** John C. Morris, Anne M. Fagan, Kaj Blennow, Henrik Zetterberg, Amanda Heslegrave, Per M. Johansson, Johan Svensson, Bengt Nellgård, Alberto Lleó, Daniel Alcolea, Jordi Clarimon, Lorena Rami, José Luis Molinuevo, Marc Suárez-Calvet, Estrella Morenas-Rodríguez, Gernot Kleinberger, Christian Haass, Michael Ewers, Johannes Levin, Martin R. Farlow, Richard J. Perrin.

**Supervision:** Carlos Cruchaga.

**Validation:** Carlos Cruchaga.

**Writing – original draft:** Muhammad Ali.

**Writing – review & editing:** Muhammad Ali, John C. Morris, Anne M. Fagan, Kaj Blennow, Henrik Zetterberg, Amanda Heslegrave, Per M. Johansson, Johan Svensson, Bengt Nellgård, Alberto Lleó, Daniel Alcolea, Jordi Clarimon, Lorena Rami, José Luis Molinuevo, Marc Suárez-Calvet, Estrella Morenas-Rodríguez, Gernot Kleinberger, Christian Haass, Michael Ewers, Johannes Levin, Martin R. Farlow, Richard J. Perrin, Carlos Cruchaga.

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
