## [Decision Letter · Decision Letter 0]

22 Mar 2022

PONE-D-22-05551Leveraging large multi-center cohorts of Alzheimer Disease endophenotypes to understand the role of Klotho heterozygosity on disease riskPLOS ONE

Dear Dr. Cruchaga,

Thank you for submitting your manuscript to PLOS ONE. After careful consideration, we feel that it has merit but does not fully meet PLOS ONE’s publication criteria as it currently stands. Therefore, we invite you to submit a revised version of the manuscript that addresses the points raised during the review process.

We look forward to receiving your revised manuscript.

Kind regards,

Kensaku Kasuga

Academic Editor

PLOS ONE

Journal Requirements:

Additional Editor Comments (if provided):

Please respond to reviewers' comments.

Reviewers' comments:

Reviewer's Responses to Questions

**Comments to the Author**

1. Is the manuscript technically sound, and do the data support the conclusions?

Reviewer #1: Yes

Reviewer #2: Yes

2. Has the statistical analysis been performed appropriately and rigorously? 

Reviewer #1: Yes

Reviewer #2: Yes

3. Have the authors made all data underlying the findings in their manuscript fully available?

Reviewer #1: No

Reviewer #2: Yes

4. Is the manuscript presented in an intelligible fashion and written in standard English?

Reviewer #1: Yes

Reviewer #2: Yes

5. Review Comments to the Author

Reviewer #1: Authors conducted a candidate-gene association study with five different AD endophenotypes, brain amyloidosis (PiB PET scan image; and n = 5,541) and CSF Ab42 (n = 5,093), Tau (n = 5,127), pTau (n = 4,778) and sTREM2 (n = 2,123). The target gene in this study was Klotho gene (KL) on chromosome 13, previously reported as a protective gene of AD. As described in the main body of the text (rows 353 – 356 on p19 in Discussion), the sample size is the largest in terms of evaluation of the genetic association of KL-VS^HET+^ with the endophenotypic data of AD. The study is well designed, and the conclusion is very clear although the functional relationship between KL and Ab/tau metabolism is not yet well understood and is an important issue to be addressed in the future. Judging from the MAF of the two single nucleotide variants, rs9536314 (p.Phe352Val) and rs9527025 (p.Cys370Ser), such studies are indeed considered valid in Caucasians: actually, the MAF (0.1 - 0.2 in gnomAD) is common, for example, compared to East Asian’s MAF (0.001 in gnomAD), making it a bit easy to find variant carriers. The fact that these two variants have a protective role against APOE-e4 allele in cognitive normal subjects aged 60-80 y.o. is indeed a promising finding although it does not pass the multiple comparison correction. Overall, there are no major revisions, but there are several minor points that need to be corrected and are listed below.

Minor revision:

1) There is no uniformity in the way papers are cited: for example, references #8, 12, 17, 35, 36, and 52. Concerning reference 56, publication date is wrong: 2014 > 1997. Please carefully check the section of References again.

2) Please correct the word "KL-VSHET+" to superscript (KL-VS^HET+^) in Table 2 (p15).

3) I wonder if a description of Figure S1 is not able to find.

4) It is helpful for readers to present representative brain imaging data (i.e., amyloid PET images) of KL-VS^HET+^ to easily understand the protective impact of the variants in the main body of the manuscript or a supplementary figure.

Reviewer #2: Ali et al. examined the association between APOEε4 and five AD endophenotypes (Amyloid PET, CSF Aβ42, CSF Tau, CSF pTau, and CSF sTREM2) in a hetero group (KL-VSHET+) with one copy of the VS haplotype, a functional haplotype of the Klotho gene, using a total of 9,500 participants. The results showed that KL-VSHET+ was significantly associated with CSF Aβ42, CSF Tau, and CSF pTau, especially in the APOEε4+ group. As the authors mentioned in the introduction, previous studies on the association between KL-VS and AD pathology have reported conflicting results, but this study supports that KL-VS is associated with AD pathology by validating it with a larger sample than previous studies. Overall, the study is well written in detail and the use and interpretation of statistical methods are reasonable.

Please respond to the following comments and minor concerns.

1. The authors mentioned in the introduction that the hetero group with one copy of KL-VS is more protective of the brain than the homo group with two copies of KL-VS (KL-VSHET-). Intuitively, I thought that having more KL-VS haplotypes would be important for brain protection, but this does not seem to be the case. Although the authors examined only the group with 1 copy of KL-VS in this study, I am also interested in the association between APOEε4 and AD pathologies in the group with 0 copies of KL-VS and the group with 2 copies of KL-VS.

2. In Table 2, the authors found a significant decrease in CSF Tau and pTau in APOEε4 carriers compared to APOEε4 non-carriers. To show more clear statistical results, I recommend that the authors perform an association analysis including an interaction effect for the presence or absence of APOE4 and the status of each biomarker (positive or negative) and include the statistics in the paper.

3. Figure S1 is not included in the text.

4. The PCA plot in Figure S3 should have a smaller plot size and be color-coded by the study as in Figure S2.

6. PLOS authors have the option to publish the peer review history of their article (what does this mean?). If published, this will include your full peer review and any attached files.

Reviewer #1: No

Reviewer #2: No

---

## [Author Response · Author response to Decision Letter 0]

31 Mar 2022

Dear Reviewers,

We are immensely thankful for your insightful comments and suggestions. We believe your suggestions will improve the quality of the manuscript, therefore, we have addressed all the comments/suggestions as per the best of our understanding and available data. For further details, please see the answers to the specific questions/comments below:

Reviewer 1:

Minor revision:

1) There is no uniformity in the way papers are cited: for example, references #8, 12, 17, 35, 36, and 52. Concerning reference 56, publication date is wrong: 2014 > 1997. Please carefully check the section of References again.

Answer: All of the references have been carefully corrected and standardized as per the format of the journal.

2) Please correct the word "KL-VSHET+" to superscript (KL-VSHET+) in Table 2.

Answer: Corresponding text has been changed in the table title (see track changes).

3) I wonder if a description of Figure S1 is not able to find.

Answer: This figure was not referenced in the manuscript text. We have created a reference to this image in line 287 on page 13 (see track changes).

4) It is helpful for readers to present representative brain imaging data (i.e., amyloid PET images) of KL-VSHET+ to easily understand the protective impact of the variants in the main body of the manuscript or a supplementary figure.

Answer: Unfortunately, we do not have access to the imaging data. We only have access to the raw mean SUVR scores for each sample; therefore, we are unable to provide this information.

 

Reviewer 2:

1. The authors mentioned in the introduction that the hetero group with one copy of KL-VS is more protective of the brain than the homo group with two copies of KL-VS (KL-VSHET-). Intuitively, I thought that having more KL-VS haplotypes would be important for brain protection, but this does not seem to be the case. Although the authors examined only the group with 1 copy of KL-VS in this study, I am also interested in the association between APOEε4 and AD pathologies in the group with 0 copies of KL-VS and the group with 2 copies of KL-VS.

Answer: We have assessed the associations between APOE4 and AD pathologies in the group with 2 and 0 copies of KL-VS haplotype (KL-VSHET-). Results showed no significant association across any AD endophenotypes (Table S5), confirming the existing hypothesis that suggests the neuroprotective effect of KL heterozygosity (KL-VSHET+) but not the homozygosity (KL-VSHET-). An additional supplementary table (Table S5) and corresponding text has been added in the manuscript (lines 305 – 313; see track changes).

2. In Table 2, the authors found a significant decrease in CSF Tau and pTau in APOEε4 carriers compared to APOEε4 non-carriers. To show more clear statistical results, I recommend that the authors perform an association analysis including an interaction effect for the presence or absence of APOE4 and the status of each biomarker (positive or negative) and include the statistics in the paper.

Answer: We have used an updated model where APOE4 interaction effect was included with the status (positive or negative) for each biomarker. The general linear model used for this analysis is:

glm(Bin.score ~ Sex + Age + PC1 + PC2 + PC3 + rs9536314*APOE4+)

Where Bin.Score represent dichotomized biomarker status and APOE4+ represents the binarized APOE4 status (0 = APOE-, 1= APOE4+). By using the described interaction effect model we did not observed any significant association between KL-VSHET+ and available endophenotypes. Text explaining these results has been added in the manuscript (line 309-313; see track changes) and a new supplementary table (Table S4) has been added. 

3. Figure S1 is not included in the text.

Answer: We have created a reference to this image in line 287 (see track changes).

4. The PCA plot in Figure S3 should have a smaller plot size and be color-coded by the study as in Figure S2.

Answer: We have created a new image in which points are color-coded by the study (Figure S3).

---

## [Decision Letter · Decision Letter 1]

6 Apr 2022

Leveraging large multi-center cohorts of Alzheimer Disease endophenotypes to understand the role of Klotho heterozygosity on disease risk

PONE-D-22-05551R1

Dear Dr. Cruchaga,

We’re pleased to inform you that your manuscript has been judged scientifically suitable for publication and will be formally accepted for publication once it meets all outstanding technical requirements.

Kind regards,

Kensaku Kasuga

Academic Editor

PLOS ONE

Additional Editor Comments (optional):

The authors fully responded to the reviewer's comments.

The manuscript deserves to be published.

Reviewers' comments:

Reviewer's Responses to Questions

**Comments to the Author**

1. If the authors have adequately addressed your comments raised in a previous round of review and you feel that this manuscript is now acceptable for publication, you may indicate that here to bypass the “Comments to the Author” section, enter your conflict of interest statement in the “Confidential to Editor” section, and submit your "Accept" recommendation.

Reviewer #1: All comments have been addressed

Reviewer #2: All comments have been addressed

2. Is the manuscript technically sound, and do the data support the conclusions?

Reviewer #1: Yes

Reviewer #2: Yes

3. Has the statistical analysis been performed appropriately and rigorously? 

Reviewer #1: Yes

Reviewer #2: Yes

4. Have the authors made all data underlying the findings in their manuscript fully available?

Reviewer #1: No

Reviewer #2: Yes

5. Is the manuscript presented in an intelligible fashion and written in standard English?

Reviewer #1: Yes

Reviewer #2: Yes

6. Review Comments to the Author

Reviewer #1: The authors have responded to all my comments/suggestions with precision.

Therefore, there are no further corrections.

Reviewer #2: The authors have addressed all of my concerns. The revised manuscript is ready for publication. I recommend publication of this manuscript in PLOS ONE.

7. PLOS authors have the option to publish the peer review history of their article (what does this mean?). If published, this will include your full peer review and any attached files.

Reviewer #1: No

Reviewer #2: No

---

## [Editor Report · Acceptance letter]

16 May 2022

PONE-D-22-05551R1 

Leveraging large multi-center cohorts of Alzheimer Disease endophenotypes to understand the role of Klotho heterozygosity on disease risk 

Dear Dr. Cruchaga:

I'm pleased to inform you that your manuscript has been deemed suitable for publication in PLOS ONE. Congratulations! Your manuscript is now with our production department. 

Kind regards, 

on behalf of

Dr. Kensaku Kasuga 

Academic Editor

PLOS ONE